# TrajTok: What makes for a good trajectory tokenizer in behavior generation?

**Zhiyuan Zhang[1], Xiaosong Jia[2,3†], Guanyu Chen[1], Qifeng Li[1],**
**Zuxuan Wu[2,3], Yu-Gang Jiang[2,3], Junchi Yan[1†]**

1. Sch. of Artificial Intelligence & Sch. of Computer Science, Shanghai Jiao Tong University
2. Institute of Trustworthy Embodied AI, Fudan University
3. Shanghai Key Laboratory of Multimodal Embodied AI
[†]Correspondence authors
https://github.com/Thinklab-SJTU/TrajTok

## Abstract

Behavior generation in autonomous driving aims to simulate dynamic driving scenarios from recorded driving logs. A popular approach is to apply next-token-prediction with discrete trajectory tokenization. In this work, we explore what makes a good trajectory tokenizer from the perspective of logged data usage. We first analyze the four properties (coverage, utilization, symmetry and robustness) of vocabularies of data-driven and rule-based trajectory tokenizers and their impact on performance and generalization. Data-driven tokenizers often build vocabularies with better utilization but suffer from insufficient coverage and sensitivity to noise, while rule-based methods have better coverage but contain too many useless tokens. With these insights, we propose TrajTok, a trajectory tokenizer that combines the two methods with rule-based vocabulary candidate setup and data-driven filtering and selection processes. The tokenizer has balanced coverage and utilization as well as good symmetry and robustness. Furthermore, we propose a spatial-aware label smoothing method for the cross-entropy loss to better model the similarities between the trajectory tokens. Our method wins first place in the 2025 Waymo Open Sim Agents Challenge.

## 1 Introduction

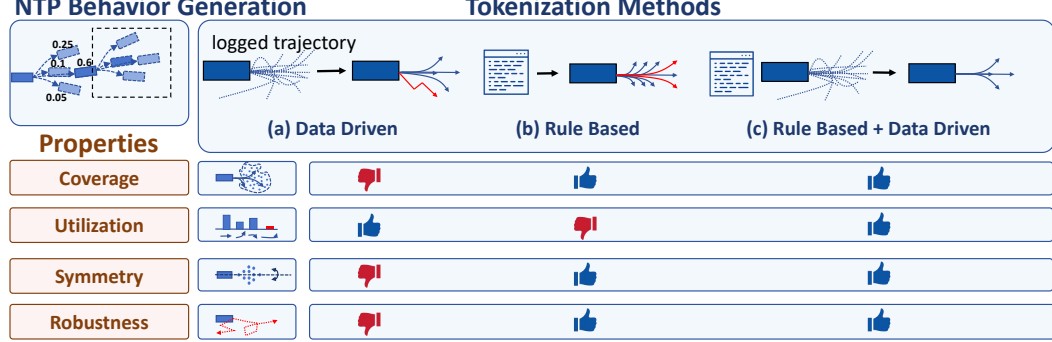

Figure 1: **Comparison of tokenizers with different logged data usage in NTP behavior generation model.** Combining data-driven and rule-based method, the proposed TrajTok balances coverage and utilization while has well symmetry and robustness.

Behavior generation aims to simulate realistic and dynamic chains of actions using logged data. In autonomous driving, behavior generation is a key component in simulators for data collection and evaluation. It typically takes the historical trajectories of agents and environmental information as inputs and generates future multi-agent trajectories auto-regressively (Chen et al., 2024).

This work was supported by Scientific Research Innovation Capability Support Project for Young Faculty (U40) of the Ministry of Education of China, SRICSPYF-ZY2025019. This work also was in part supported by the Science and Technology Commission of Shanghai Municipality (No. 24511103100) and the New Cornerstone Science Foundation through the XPLORER PRIZE.

Recently, inspired by large language models (LLMs) (Floridi & Chiriatti, 2020; Touvron et al., 2023; Yang et al., 2023), a series of behavior generation models adopt the next-token-prediction (NTP) paradigm (Philion et al., 2023; Seff et al., 2023; Wu et al., 2024; Zhao et al., 2024), as shown in Fig. 1 (upper left). These models build a vocabulary, a finite set of trajectories that tries to model the continuous distribution space with trajectory tokenizers. With the trajectory vocabulary, these models transform the complex trajectory distribution in continuous space into a simpler discrete space for easier multi-modal behavior handling and model fitting.

Due to the low-dimensional nature of trajectories, various methods can be applied for trajectory tokenization, including data-driven methods such as VQ-VAE (Van Den Oord et al., 2017), K-means (Arthur & Vassilvitskii, 2006) and K-disks (Philion et al., 2023), and rule-based methods such as gridding (Seff et al., 2023). However, which method and what properties lead to a good trajectory tokenizer remain unclear to the community.

In this paper, we study what makes a good trajectory tokenizer and how to use the logged data in the tokenization process. First, we analyze the properties of trajectory tokens, including coverage, utilization, symmetry, and robustness, as well as their impact on the performance and generalization of NTP behavior generation models. As shown in Fig. 1 (a), we find that data-driven tokenizers tend to produce vocabularies within the recorded distribution, resulting in high utilization but low coverage. Additionally, their sensitivity to noisy data results in erroneous vocabulary and the asymmetry affects generalization. In contrast, rule-based methods cover a wider region with good symmetry and robustness, which leads to better generalization. However, they rely solely on human priors without adapting to the real data distribution, resulting in vocabularies that contain many redundant trajectories that do not appear in reality and low utilization, as shown in Fig. 1 (b). Under the same vocabulary size, their performance is far inferior to data-driven methods.

Based on these analyses, we propose TrajTok, a tokenizer specifically designed for trajectories that combines data-driven and rule-based methods, as shown in Fig. 1 (c). It first sets up a primary vocabulary candidate with rules, then applies data-driven filtering and expansion. The former ensures the stability and symmetry of the vocabulary, while the latter balances coverage and utilization.

Furthermore, we study loss designs related to vocabularies. A few NTP models use cross-entropy loss with label smoothing (Szegedy et al., 2016) to reduce overfitting and improve generalization. The standard label smoothing mechanism assigns the same weight to all non-ground-truth tokens, ignoring their similarity to ground-truth ones. As the similarity of trajectories is closely related to spatial distance, we propose a spatial-aware label smoothing method that assigns different weights to tokens conditioned on their distances to the ground-truth one. This smoothing method improves the performance of NTP behavior generation models effectively.

Our method wins first place in the Waymo Open Sim Agent Challenge (WOSAC) (Montali et al., 2023) 2025 with a Realism Meta of 0.7852, with good generalization across datasets.

Our contributions are threefold:

- We conduct a detailed investigation of trajectory tokenization in NTP behavior generation from the perspective of logged data usage and analyze the coverage, utilization, symmetry, and robustness of data-driven and rule-based methods.

- We propose TrajTok, a plug-and-play trajectory tokenizer that combines rule-based and data-driven methods, which achieves state-of-the-art performance on the Waymo Open Sim Agent Challenge.

- We explore token-related loss design in NTP behavior generation models and propose a spatial-aware label smoothing method for the cross-entropy loss.

## 2 RELATED WORK

### 2.1 BEHAVIOR GENERATION MODELS

Behavior generation modules are key components in simulators and benchmarks (Jia et al., 2024b; You et al., 2024; Zhou et al., 2025; Jia et al., 2021) to evaluate the ability of autonomous driving models (Hu et al., 2023; Lu et al., 2024; Jia et al., 2025a; 2023d; Wu et al., 2022) and serve as

environments for reinforcement learning (Li et al., 2025; Jia et al., 2023b; Yang et al., 2025c). The task proposed by Waymo Open Sim Agent Challenge (WOSAC) (Montali et al., 2023) aims to generate next states (including position and heading) of agents step by step realistically conditioned on historical steps and environmental information. The traditional encoder-decoder architectures (Shi et al., 2022; Zhou et al., 2023; Jia et al., 2023c; 2022) in motion prediction can be used in this task, but they suffer from low data utilization and significant out-of-distribution (OOD) problems in the auto-regressive generation process (Zhou et al., 2024; Jia et al., 2023a). Inspired by large language models (LLM) and vision-language-action (VLA) Models (Yang et al., 2025b; Fan et al., 2025), most recent works adapt the next-token-prediction (NTP) paradigm (Jia et al., 2024a). Trajeglish (Philion et al., 2023) first models the behavior generation task as discrete NTP with a data-driven tokenizer K-disks. SMART (Wu et al., 2024) introduces a map discrete tokenizer and further analyzes the scalability of the architecture, while KiGRAS (Zhao et al., 2024) factorizes the driving scene in action space with kinematic transformations. Furthermore, CATK (Zhang et al., 2025) introduces a closed-loop fine-tuning strategy to further address the OOD problem.

Despite NTP-based models, diffusion models are another approach for behavior generation. These models focus on generating controllable (Zhong et al., 2022; Huang et al., 2024; Jiang et al., 2024; Pronovost et al., 2023; Yang et al., 2025a) or adversarial (Xie et al., 2024; Yin et al., 2024; Chang et al., 2024; Xu et al., 2025) behaviors with guidance, while their realism is much lower than NTP-based models on WOSAC.

## 2.2 TRAJECTORY TOKENIZERS FOR BEHAVIOR GENERATION

Tokenizers are widely used in deep-learning, such as getting image features in perception tasks (Li et al., 2023; Jia et al., 2020; Wu et al., 2023; Gao et al., 2020; Zhu et al., 2025; Jia et al., 2025b). In behavior generation, trajectory tokenizers transform the states of agents from continuous space to discrete space. Various tokenizers are adopted in discrete NTP behavior generation models, including VQ-VAE, K-means, K-disks and naive grid-based methods. Most of them are long-existing general approaches and only the K-disk proposed by Trajeglish (Philion et al., 2023) is specifically designed for trajectory tokens. K-disks randomly select a state in logged data as a trajectory token and exclude all logged states within a certain distance from the selected states, then randomly select the next token in the remaining data. Therefore, it is more like a sampling algorithm rather than a clustering algorithm. Grid-based methods such as that used in MotionLM (Seff et al., 2023) simply select the token with a uniform distribution in a pre-defined region of each state component.

Varying from deep-learning methods, clustering or sampling algorithms and rule-based methods, these tokenizers rely on logged trajectory data at different levels. However, there is a lack of comprehensive analysis of these tokenizers and the data usage preference behind them. Trajeglish (Philion et al., 2023) compares some tokenizers only in terms of discretization error, without studying their impacts on the performance of behavior generation models.

## 3 METHODS

### 3.1 PROBLEM FORMULATION

The behavior generation task can be defined as follows: Given an initial scene, including the HD map $\mathcal{M}$ and the past $T_h$ states of all agents $\{\mathcal{S}_{-T_h}, ..., \mathcal{S}_0\}$, the goal is to generate the states of all agents at each future time step within $T_f$ steps, i.e., $\{\mathcal{S}_1, ..., \mathcal{S}_{T_f}\}$.

We focus on the discrete NTP behavior generation models that output trajectories as actions, such as SMART (Wu et al., 2024) and Trajeglish (Philion et al., 2023). We denote the model as $\mathcal{N}_{\theta,V}$ with trainable parameters $\theta$ and trajectory vocabulary $\mathcal{V} = \{c_1, c_2, ..., c_{|\mathcal{V}|}\}$. Each trajectory token $c_i \in \mathbb{R}^{L \times 3}$ consists of $L$ points with (x, y, yaw) in the agent-centric coordinate. We denote the number of all agents as $N_A$, and the model outputs the trajectory tokens $a_t \in \mathbb{R}^{N_A \times L \times 3}$ for each agent in the interval $L$ as:

$$a_t = \mathcal{N}_{\theta,\mathcal{V}}(\mathcal{S}_{-T_h:t \times L}, \mathcal{M}) \tag{1}$$

where $t \in \{0, 1, 2, ..., (T_f//L)\}$ is the end timestamp for each interval. The output trajectory token for each agent is selected from the vocabulary, i.e. $a_t^{(i)} \in \mathcal{V}$. Then, a coordinate transform $f$ is applied to calculate states in the next interval:

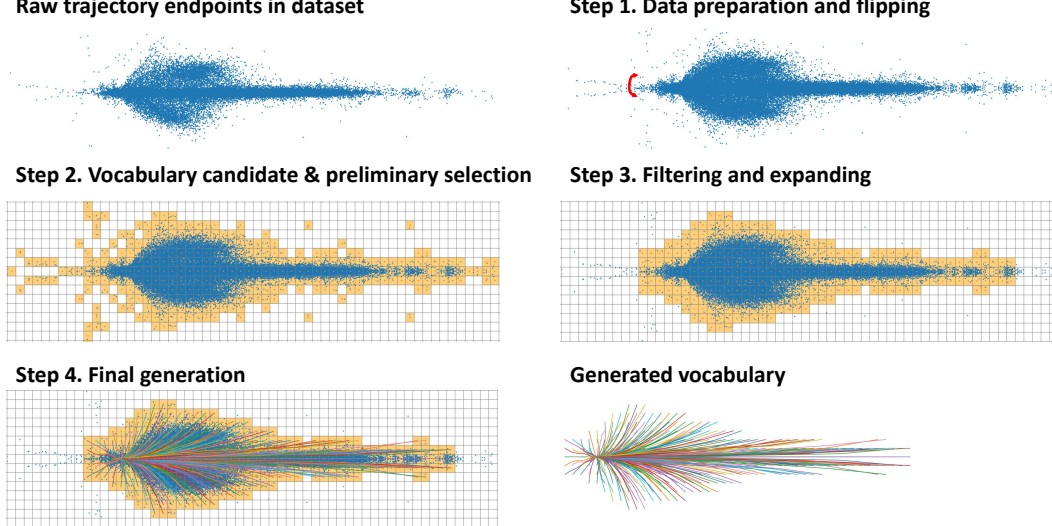

Figure 2: **The vocabulary generation process of TrajTok.** The blue points are endpoints of logged trajectories and the yellow grids are selected in each process.

$$\mathcal{S}_{t \times L+1:t \times (L+1)+1} = f(a_t, \mathcal{S}_t) \tag{2}$$

The tokenizer $\mathrm{Tok}()$ generates the vocabulary $\mathcal{V}$. Pure rule-based tokenizers only use predefined hyper-parameters $\theta_{tok}$, such as the vocabulary size and sample range as input, i.e.

$$\mathcal{V}_{\text{rule\_based}} = \mathrm{Tok}(\theta_{\text{tok}}) \tag{3}$$

Data-driven tokenizers generate the vocabulary with a dataset $\mathcal{D}$, i.e.

$$\mathcal{V}_{\text{data\_driven}} = \mathrm{Tok}(\theta_{\text{tok}}, \mathcal{D}) \tag{4}$$

### 3.2 TRAJTOK

As Fig. 2 shows, TrajTok generates the trajectory vocabulary through the following steps:

**Step 1: Data preparation and flipping.** First, all valid trajectories of length L in the dataset are extracted and normalized to the agent-centric coordinate system. We denote the normalized trajectories as $D \in \mathbb{R}^{N_D \times L \times 3}$, where $N_D$ is the number of trajectories. Then, the trajectories are flipped along the x-axis and concatenated with original trajectories, i.e.

$$\widetilde{D} = \mathrm{Concatenate}(D, \mathrm{Flip}(D)) \tag{5}$$

The flipping operation ensures the symmetry of trajectories, which results in symmetric tokens along the x-axis.

**Step 2: Rule-based vocabulary candidate setup and preliminary selection.** The vocabulary candidate is set up with a grid. Given ranges $x_{\min}, x_{\max}, y_{\min}, y_{\max}$ and intervals $x_{\text{interval}}, y_{\text{interval}}$, the size of the grid is $(H, W) = \left( \frac{y_{\max} - y_{\min}}{y_{\text{interval}}}, \frac{x_{\max} - x_{\min}}{x_{\text{interval}}} \right)$. Each trajectory in flipped data $\widetilde{D}$ is associated with a cell if its endpoint falls within the cell. We denote $\hat{D}_{ij} \in \mathbb{R}^{N_{ij}^{\text{traj}} \times L \times 3}$ as all trajectories associated with cell $(i, j)$ and $N_{ij}^{\text{traj}}$ as the number of these trajectories for each grid. For preliminary selection, a grid is marked as valid if $N_{ij}^{\text{traj}}$ is larger than a threshold $s_p$. The validity binary map $B$ is set through

$$B_{ij} = \mathbb{1}_{[N_{ij}^{\text{traj}} \geq s_p]} \tag{6}$$

**Step 3: Data-driven filtering and expanding.** The preliminary selected grids are within the distribution of logged data, and are sensitive to noise, as shown in Fig 2. To cover more possible trajectories in reality and improve robustness, filtering and expanding are then applied. For a grid $(i, j)$ on the map, we compute the number of its selected neighbors within the distance of k, i.e.

$$N_{ij}^{\text{vb}} = \sum_{m=i-k}^{i+k} \sum_{n=j-k}^{j+k} B_{mn} \tag{7}$$

Then we add an unselected grid from step 2 if the number of its selected neighbors $N_{ij}^{\text{vb}}$ is larger than the threshold $s_a$, and remove a selected grid if $N_{ij}^{\text{vb}}$ is less than the threshold $s_r$ through

$$\hat{B}_{ij} = \begin{cases} 1 & \text{if} \quad B_{ij} = 0 \quad \text{and} \quad N_{ij}^{\text{vb}} \geq s_a \\ 0 & \text{if} \quad B_{ij} = 1 \quad \text{and} \quad N_{ij}^{\text{vb}} \leq s_r \end{cases} \tag{8}$$

**Step 4: Final generation.** Finally, the trajectory vocabulary is generated for each selected grid. The average of logged data trajectories is used if there are trajectories associated with the cell.

$$V_1 = \left\{ \frac{\sum_{m=0}^{N_{ij}^{\text{traj}}} \hat{D}_{ijm}}{N_{ij}^{\text{traj}}} \quad | \quad i \in [0, W] \wedge j \in [0, H] \wedge \hat{B}_{ij} = 1 \wedge N_{ij}^{\text{traj}} > 0 \right\} \tag{9}$$

If not, the vocabulary is generated with curve interpolation from the origin point to the center of the grid $p_{ij}$. This happens if the cell is set valid in the expanding process. The yaw of the endpoint $r_{ij}$ is estimated from the trajectories in nearby grids.

$$V_2 = \{ \text{Curve\_Interp}(0, p_{ij}, r_{ij}, L) \quad | \quad i \in [0, W] \wedge j \in [0, H] \wedge \hat{B}_{ij} = 1 \wedge N_{ij}^{\text{traj}} = 0 \} \tag{10}$$

The final vocabulary is:

$$V_{\text{TrajTok}} = V_1 \cup V_2 \tag{11}$$

### 3.3 SPATIAL-AWARE LABEL SMOOTHING

NTP models often use cross-entropy loss with label smoothing (Szegedy et al., 2016) to alleviate overfitting and improve generalization. Standard label smoothing assigns the same probability to each non-ground-truth label. Denoting the index of the ground-truth label as $j$ and the parameter of label smoothing as $\varepsilon$, the target probability $y$ for each label $i$ is calculated as:

$$y_i = \begin{cases} 1 - \varepsilon & \text{if} \quad i = j \\ \frac{\varepsilon}{|\mathcal{V}|} & \text{if} \quad i \neq j \end{cases} \tag{12}$$

Under this mechanism, the loss is the same for predicting a trajectory token from the ground truth and one next to the ground truth. However, the former impacts the performance of behavior generation more significantly. Thus, we hope the model can be more tolerant of tokens that are spatially close to the ground-truth, while rejecting tokens that are far away. We calculate the average error between each token trajectory and the ground-truth trajectory, and assign the target probability inversely proportional to the square of the error:

$$k_i = \frac{1}{||\mathbf{c_i} - \mathbf{c_j}||^2} \tag{13}$$

$$y_i = \begin{cases} 1 - \varepsilon & \text{if} \quad i = j \\ \frac{\varepsilon k_i}{\sum_{m=0, m \neq j}^{|\mathcal{V}|} k_m} & \text{if} \quad i \neq j \end{cases} \tag{14}$$

This spatial-aware label smoothing can both reduce errors and improve generalization.

## 4 EXPERIMENTS

### 4.1 EXPERIMENTAL SETUP

**Dataset.** We conduct experiments on the Waymo Open Motion Dataset (WOMD) (Ettinger et al., 2021). There are 486,995/44,097/44,920 scenarios in the training/validation/test set. Each scenario contains 9 seconds of agent states with an interval of 0.1s and a high-definition map. The behavior generation task is to generate the states in the future 8 seconds of up to 128 agents with states in the past 1 second in an autoregressive manner.

**Metrics.** We use the metrics in the Waymo Open Sim Agents Challenge (Montali et al., 2023). The main metric is the Realism Meta, which is composed of the Kinematic, Interactive, and Map-based

Table 1: **Waymo Open Sim Agents Challenge leaderboard 2025.** Top 15 entries in the Submission Period are presented. ▆ highlights the best metric and ▆ highlights the second best.

| Method | Realism Meta ↑ | Kinematic↑ | Interactive ↑ | Map-based ↑ | minADE ↓ |
|---|---|---|---|---|---|
| SMART-R1 | 0.7855 | 0.4940 | 0.8109 | 0.9194 | 1.2990 |
| TrajTok (Ours) | 0.7852 | 0.4887 | 0.8116 | 0.9207 | 1.3179 |
| unimotion | 0.7851 | 0.4943 | 0.8105 | 0.9187 | 1.3036 |
| SMART-tiny-CLSFT (Zhang et al., 2025) | 0.7846 | 0.4931 | 0.8106 | 0.9177 | 1.3065 |
| SMART-tiny-RLFTSim | 0.7844 | 0.4893 | 0.8128 | 0.9164 | 1.3470 |
| comBOT | 0.7837 | 0.4899 | 0.8102 | 0.9175 | 1.3687 |
| AgentFormer | 0.7836 | 0.4906 | 0.8103 | 0.9167 | 1.3422 |
| UniMM (Lin et al., 2025) | 0.7829 | 0.4914 | 0.8089 | 0.9161 | 1.2949 |
| R1Sim | 0.7827 | 0.4894 | 0.8105 | 0.9147 | 1.3593 |
| SimFormer | 0.7820 | 0.4920 | 0.8060 | 0.9167 | 1.3221 |
| SMART-tiny-RLFT | 0.7815 | 0.4853 | 0.8107 | 0.9133 | 1.4266 |
| SMART_topk32 | 0.7814 | 0.4854 | 0.8089 | 0.9153 | 1.3931 |
| SMART-tiny-RLFT | 0.7780 | 0.4799 | 0.8070 | 0.9109 | 1.6388 |
| llm2ad | 0.7779 | 0.4846 | 0.8048 | 0.9109 | 1.2827 |
| UniTFormer | 0.7776 | 0.4892 | 0.7997 | 0.9140 | 1.3592 |

Table 2: **Performance of SMART model with different tokenizers on validation split of WOMD.** The models are trained on a random 20% subset of the training set and the metrics are of WOSAC 2024 version. The tokenizer of original SMART model is K-disks.

| Tokenizer | Realism Meta ↑ | Kinematic↑ | Interactive ↑ | Map-based ↑ | minADE ↓ |
|---|---|---|---|---|---|
| VQ-VAE | 0.7596 | 0.4629 | 0.8101 | 0.8642 | 1.3982 |
| K-means | 0.7476 | 0.4375 | 0.7903 | 0.8635 | 1.4797 |
| K-disks | 0.7584 | 0.4602 | 0.8004 | 0.8748 | 1.3532 |
| Grid | 0.7527 | 0.4121 | 0.8099 | 0.8737 | 1.4137 |
| TrajTok | 0.7702 | 0.4867 | 0.8132 | 0.8769 | 1.3428 |

Table 3: **Performance of K-disks and TrajTok across datasets.**

| Tokenizer | Logged Dataset | Realism Meta ↑ | minADE ↓ |
|---|---|---|---|
| K-disks | Waymo | 0.7584 | 1.3537 |
| K-disks | nuScenes | 0.7350 (-0.0234) | 1.4074 |
| TrajTok | Waymo | 0.7702 | 1.3428 |
| TrajTok | nuScenes | 0.7641 (-0.0061) | 1.3681 |

Table 4: **Performance of K-disks and TrajTok with different sizes of logged data.**

| Tokenizer | Logged Data Size | Realism Meta ↑ | minADE ↓ |
|---|---|---|---|
| K-disks | $10^7$ | 0.7584 | 1.3537 |
| K-disks | $10^6$ | 0.7539 (-0.0045) | 1.3628 |
| K-disks | $10^5$ | 0.7442 (-0.0142) | 1.3696 |
| TrajTok | $10^7$ | 0.7702 | 1.3428 |
| TrajTok | $10^6$ | 0.7691 (-0.0011) | 1.3445 |
| TrajTok | $10^5$ | 0.7675 (-0.0027) | 1.3511 |

metrics. Notably, it takes several days to compute the metrics on the whole validation set with the official code. Previous works (Zhang et al., 2025) conduct ablation studies with evaluation on a small subset, which may lead to biased results. We improve the code and decrease the computation time to less than 2 hours while strictly aligning the protocol and results with the official version. With the efficient evaluation tool, all our ablation studies are evaluated on the whole validation set.

## 4.2 RESULTS

**WOSAC leaderboard.** Table 1 lists the results of the top 15 entries in the 2025 Waymo Open Sim Agents Challenge. TrajTok wins the first place in the Waymo Open Sim Agent Challenge 2025 with a Realism Meta metric of 0.7852. It also reaches state-of-the-art performance on Map-based metrics of 0.9207 and competitive performance on other metrics. Without any fine-tuning processes, our approach has comparable or superior performance compared to other methods that are also developed on the SMART-tiny model but need fine-tuning, such as SMART-tiny-CLSFT (Zhang et al., 2025).

**Comparison with other tokenizers.** We compare TrajTok with other tokenizers, including VQ-VAE, K-means, K-disks and grid method on the SMART (Wu et al., 2024) model, as shown in Table 2. Results show that our method reaches the best performance among all tokenizers that are adopted in behavior generation models.

**Generalization.** The generalization of the tokenizer is reflected in its ability to handle diverse real-world data with vocabularies derived from any data record. We investigate generalization of TrajTok through two experimental setups: ❶ Building vocabularies with nuScenes (Caesar et al.,

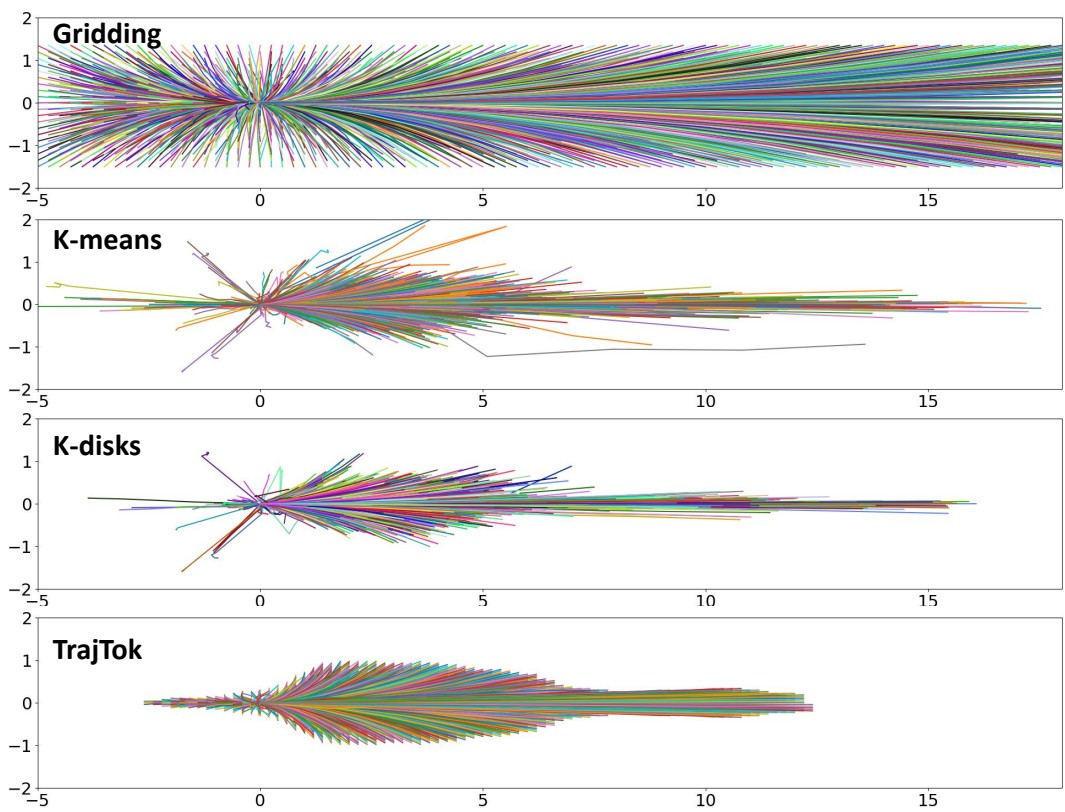

Figure 3: **Trajectory vocabularies from TrajTok and other tokenizers.** Each colored line represents a trajectory token within 0.5 seconds in agent-centric coordinate system. The size of all four vocabularies is 2000.

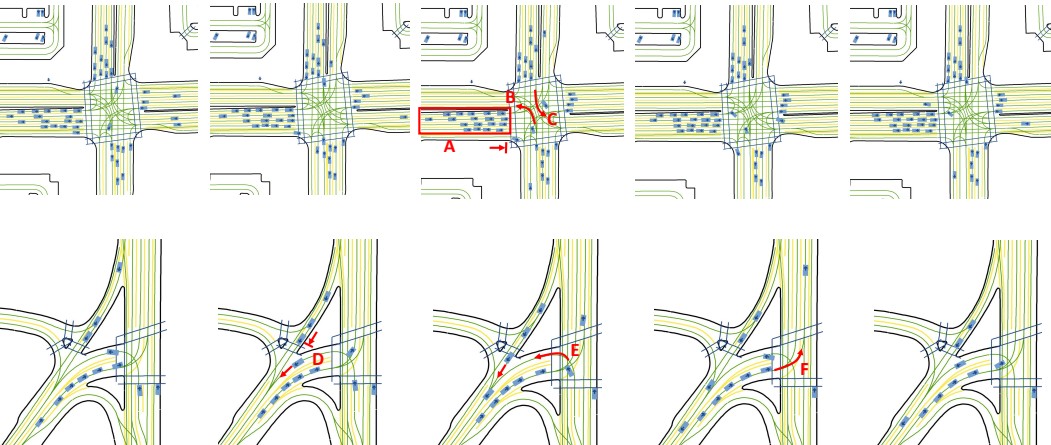

Figure 4: **Generated behavior in two scenarios.**

2020) data and training and evaluating on WOMD to evaluate the tokenizer's generalization across datasets. The results are shown in Table. 3. ❷ Building vocabularies with a small subset of WOMD to evaluate the tokenizer's ability to infer the overall distribution from few sampled data. The results are shown in Table. 4.

**Qualitative results.** Fig. 3 shows the vehicle trajectory vocabularies generated by different tokenizers. The data-driven methods K-means and K-disks produce trajectory tokens that are asymmetric and contain kinematically implausible noisy trajectories. The purely rule-based grid method generates many invalid trajectories (e.g., lateral shifts of over 1 meter within 0.5 seconds) and has lower density at the same vocabulary size. In contrast, TrajTok demonstrates good symmetry and stability, with generated trajectories that largely adhere to kinematic principles. Fig. 4 illustrates

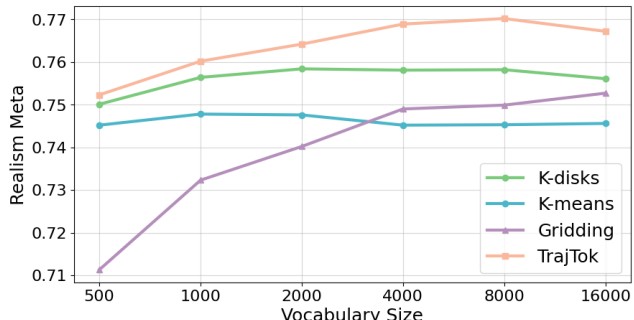

Figure 5: **Ablation of vocabulary size of vehicles.**

Table 5: **Ablation of Spatial-Aware label smoothing.**

| Tokenizer | Label Smoothing Type | Realism Meta↑ | minADE ↓ |
|-----------|---------------------|---------------|----------|
| K-disks | default | 0.7443 | 1.4230 |
| K-disks | spatial-aware | 0.7584 | 1.3537 |
| TrajTok | default | 0.7597 | 1.3797 |
| TrajTok | spatial-aware | 0.7702 | 1.3428 |

the behavior generated by TrajTok in two scenarios. The first is a busy intersection scene where traffic in direction A is queued, maintaining appropriate distances between vehicles, while vehicles in directions B and C make left turns. This demonstrates TrajTok's capability to handle interactions among a large number of vehicles precisely. The second scenario features an atypical intersection where vehicles in direction E need to turn left and backward, alternating passage with vehicles from directions D and F. Benefiting from its superior coverage, TrajTok effectively manages vehicle interactions in uncommon scenarios.

## 4.3 ABLATION

**Vocabulary Size.** We ablate the vocabulary size in Fig. 5. Increasing the vocabulary size improves the ability to represent complex distributions but may lead to model underfitting, which is also observed in LLMs (Tao et al., 2024). Rule-based gridding methods require a larger vocabulary size to reach optimal performance, while data-driven methods need a smaller size. The optimal vocabulary size for TrajTok lies between data-driven and rule-based methods. Additionally, TrajTok achieves the best performance across different vocabulary sizes.

**Spatial-Aware label smoothing.** Table 5 presents the ablation study for Spatial-Aware Label Smoothing. Whether using K-disks or TrajTok as tokenizers, Spatial-Aware Label Smoothing improves performance compared to standard label smoothing.

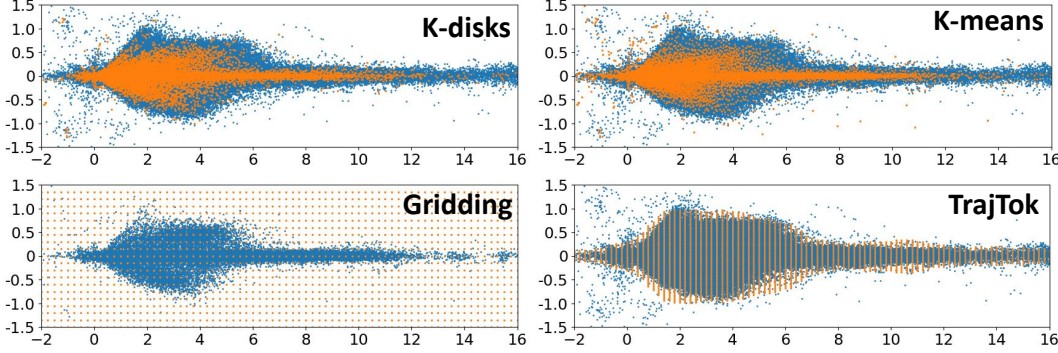

Figure 6: **The distribution of logged trajectories and vocabularies of vehicles.** Orange dots indicate the endpoints of logged trajectories, while blue dots indicate those of vocabularies.

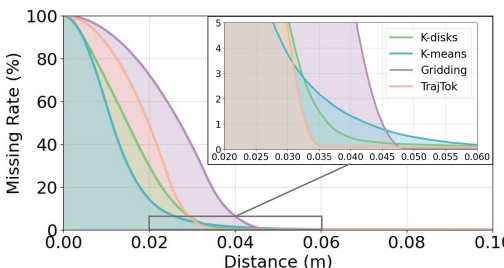

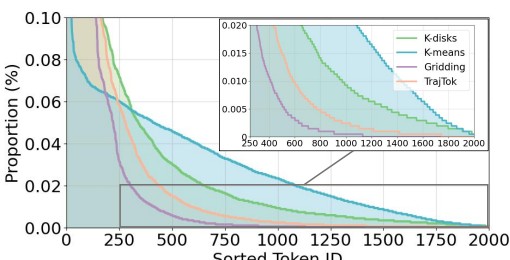

Figure 7: **The missing rate of tokenization for cyclists.**

Figure 8: **The frequency of vocabularies for cyclists.**

Table 6: **Symmetric tokenizers have better performance.**

| Tokenizer | Realism Meta↑ | minADE ↓ |
|---|---|---|
| K-disks w/ symmetry | 0.7610 | 1.3541 |
| K-disks w/o symmetry | 0.7584 | 1.3532 |
| K-means w/ symmetry | 0.7526 | 1.3519 |
| K-means w/o symmetry | 0.7476 | 1.3597 |
| TrajTok w/ symmetry | 0.7702 | 1.3428 |
| TrajTok w/o symmetry | 0.7670 | 1.3611 |

Table 7: **The average discretization error of tokenizers for vehicles.** The vocabulary size is 2000.

| Tokenizer | Realism Meta | average discretization error (m) |
|---|---|---|
| K-means | 0.7476 | 0.0224 |
| K-disks | 0.7584 | 0.0204 |
| Grid | 0.7527 | 0.0813 |
| TrajTok | 0.7702 | 0.0520 |

## 4.4 ANALYSIS AND DISCUSSIONS

**Coverage.** Qualitatively, Fig. 6 plots the endpoints of real trajectories and vocabulary trajectories. Data-driven methods generate vocabularies within the distribution of real trajectories and leave several insufficiently covered regions. The vocabulary generated by rule-based methods extends far beyond the real distribution range. In contrast, the vocabulary of TrajTok covers most of the valued trajectories. Quantitatively, Fig. 7 shows the missing rate of tokenization at different distances, which is defined as the proportion of trajectories with discretization errors larger than a given distance. TrajTok has a better missing rate at larger distances. It shows that TrajTok has appropriate coverage and is better at dealing with the long tail effect in tokenization.

**Utilization.** Fig. 8 shows the frequency of each token in the vocabulary. Rule-based gridding methods have a large number of tokens with zero frequency. Although the frequency of uncommon tokens of TrajTok is lower than that of data-driven methods, most of them match with several logged trajectories. With the increase of tokens for rare trajectories, there is a trade-off between coverage and utilization. TrajTok balances these two properties.

**Symmetry.** The symmetry of the possible distribution boundary of trajectories in reality comes from that of vehicle kinematic models. If a trajectory is recorded, its flipped counterpart is also physically drivable. Considering complex driving scenarios and road structures in reality, the flipped trajectory may possibly be driven in a scenario that was not logged in the dataset. In the experiments, we remove the flipping operation in the TrajTok generation process, resulting in a performance decline. Data-driven methods, due to the randomness of data distribution and algorithms, can not guarantee the generation of a symmetric vocabulary. We flip the trajectories below the x-axis to above the x-axis, generate a vocabulary of half size on the flipped trajectories, and then flip it back to obtain the complete vocabulary. The vocabularies generated in this manner show improved performance, as shown in Table 6. Both aspects of the experiments demonstrate the importance of symmetry.

**Robustness.** Fig. 3 and Fig. 6 show that TrajTok is less sensitive to noise. In contrast, K-disks may directly add noisy trajectories to the vocabulary during random sampling. K-means may also pick outliers since they have large distances to other samples.

**Does lower average discretization error lead to a better tokenizer?** Average discretization error is used to evaluate the performance of a tokenizer in previous work (Philion et al., 2023). However, as shown in Table 7, the average discretization error of K-disks is lower than that of TrajTok, yet

its performance is inferior to TrajTok. The indicator cannot entirely reflect the distance between the vocabulary and distribution in the real world. For example, it cannot well indicate the long-tail effect in Fig. 7. When the average discretization error is already low, the performance of the tokenizer is influenced mostly by other factors.

## 5 CONCLUSION

In this paper, we analyze data-driven and rule-based tokenizers from four perspectives including coverage, utilization, symmetry, and robustness. Based on the analysis, we combine data-driven and rule-based methods and introduce TrajTok. Additionally, we propose a spatial-aware label smoothing method to better model the similarities between the trajectory tokens. Experiments demonstrate the effectiveness of our methods.

## 6 ETHICS STATEMENT

The research conducted in the paper conforms with the ICLR Code of Ethics.

## 7 REPRODUCIBILITY STATEMENT

We describe the proposed tokenizer and loss in Sec. 3 and implementation details in Appendix A. The datasets are public accessible. The code will be open-sourced for reproduction.

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

## A  Implementation Details

**Model Settings.** We use the SMART-tiny (Wu et al., 2024) model as base NTP behavior generation model for experiments with TrajTok. Follow original settings, the the number of decoder layers (including Map-Agent Cross-Attention Layer, Agent Interaction Self-Attention Layer and Temporal Self-Attention Layer) is 6 and the hidden dim is 128. The interval L is set to 5 and the re-plan frequency is 2 Hz, which means the model predict the 0.5-second trajectory at 10Hz every 0.5s. The original model build different trajectory vocabularies for each type of agents (vehicle, bicycle and pedestrian) but use the same head to predict classification logits for all types. We use separate

prediction heads instead and set their output dim the same as the vocabulary size each. For spatial-aware label smoothing, the total target probability of all non-ground-truth labels $\varepsilon$ is 0.1, which is the same as standard label smoothing used in original model.

Table 8: **Detailed hyper-parameters of the submit version of TrajTok.** The units for $x_{\min}$, $x_{\max}$, $x_{\text{interval}}$, $y_{\min}$, $y_{\max}$, $y_{\text{interval}}$ are all meters.

| Agent Type | $x_{\min}$ | $x_{\max}$ | $x_{\text{interval}}$ | $y_{\min}$ | $y_{\max}$ | $y_{\text{interval}}$ | $k$ | $s_p$ | $s_a$ | $s_r$ |
|---|---|---|---|---|---|---|---|---|---|---|
| Vehicle | -5 | 20 | 0.1 | -1.5 | 1.5 | 0.05 | 4 | 1 | 20 | 20 |
| Bicycle | -1 | 8 | 0.05 | -1 | 1 | 0.05 | 4 | 1 | 20 | 20 |
| Pedestrian | -1.5 | 4.5 | 0.05 | -2 | 2 | 0.05 | 4 | 1 | 20 | 20 |

**Tokenizer.** For TrajTok, we set the grid range and interval for each type of agents as Table 8. We extract trajectories that last 0.5s from the Waymo Open Motion Dataset(WOMD). In the submit version, the sizes of vocabularies for vehicle, bicycle and pedestrian are 8040, 2798, 3001 separately.

**Training Details.** We train the model with 8×A100 80GB GPUs for 32 epochs on the training split of the WOMD with the AdamW optimizer. The total batch size is 48. The initial learning rate to $5 \times 10^{-4}$ and is decayed to $5 \times 10^{-6}$ based on the cosine annealing schedule.

## B    SENSITIVITY ANALYSIS

The sensitivity analysis on grid range, grid resolution and filtering and expansion parameters of TrajTok are shown in Table 9, Table 10 and Table 11.

Table 9: **Sensitivity analysis on grid range.**

| Vehicle | | | | Bicycle | | | | Pedestrain | | | | RealismMeta ↑ |
|---|---|---|---|---|---|---|---|---|---|---|---|---|
| $x_{\min}$ | $x_{\max}$ | $y_{\min}$ | $y_{\max}$ | $x_{\min}$ | $x_{\max}$ | $y_{\min}$ | $y_{\max}$ | $x_{\min}$ | $x_{\max}$ | $y_{\min}$ | $y_{\max}$ | |
| -5 | 20 | -1.5 | 1.5 | -1 | 8 | -1 | 1 | -1.5 | 4.5 | -2 | 2 | 0.7702 |
| -10 | 25 | -2 | 2 | -3 | 12 | -2 | 2 | -3 | 6 | -3 | 3 | 0.7701 |
| -2 | 15 | -1 | 1 | -1 | 5 | -0.75 | 0.75 | -0.5 | 3 | -1.5 | 1.5 | 0.7689 |

Table 10: **Sensitivity analysis on grid resolution.**

| Vehicle | | Bicycle | | Pedestrain | | RealismMeta ↑ |
|---|---|---|---|---|---|---|
| $x_{\text{interval}}$ | $y_{\text{interval}}$ | $x_{\text{interval}}$ | $y_{\text{interval}}$ | $x_{\text{interval}}$ | $y_{\text{interval}}$ | |
| 0.1 | 0.05 | 0.05 | 0.05 | 0.05 | 0.05 | 0.7702 |
| 0.05 | 0.05 | 0.025 | 0.025 | 0.025 | 0.025 | 0.7681 |
| 0.1 | 0.1 | 0.1 | 0.1 | 0.1 | 0.1 | 0.7687 |

## C    COMPUTATIONAL COST

We report the average inference time with different vocabulary sizes in Table 12 and training time with spatially aware label smoothing in Table 13. Larger vocabularies increase the inference time, but the cost remains relatively acceptable. Spatially aware label smoothing does not significantly increase the total computational cost.

## D    DISCUSS ON FAILURE CASES

There are two types of failure cases:

(a) Unrealistic motion patterns may be introduced if there are too many noisy logged trajectories in a certain region. With advances in data collection and annotation, as well as more precise sensors and improved annotation workflows, the occurrence of such cases can be reduced.

Table 11: **Sensitivity analysis on filtering and expansion parameters.** These parameters are the same for all agent types.

| $k$ | $s_p$ | $s_a$ | $s_r$ | RealismMeta ↑ |
|---|---|---|---|---|
| 4 | 1 | 20 | 20 | 0.7702 |
| 3 | 1 | 8 | 10 | 0.7693 |
| 4 | 4 | 20 | 20 | 0.7695 |

Table 12: **Average inference time on a single NVIDIA A100 GPU for a single rollout of 16 steps on a scenario with different vocabulary sizes.**

| Vocabulary size for all agent types | inference time |
|---|---|
| 2000 | 399 ms |
| 4000 | 421 ms |
| 8000 | 441 ms |

Table 13: **Training time for a single epoch on 8 NVIDIA A100 GPUs with a 20% training split.**

| Label smoothing type | training time |
|---|---|
| Standard | 18m 25s |
| Spatially Aware | 18m 34s |

(b) Long-tail behaviors may not be captured if the logged dataset is too small or lacks sufficient diversity. As larger and more diverse datasets become available and data collection costs decrease, these cases are expected to be reduced.

## E    MORE VISUALIZATIONS

Two extra figures are presented to illustrate how tokenization affects behavior generation. The red dots in figures represent endpoints of trajectories in the vocabulary, and the red arrows show the selected next actions.

Fig. 9 shows a vehicle navigating a dedicated right-turn lane. The K-disk vocabulary has insufficient coverage and offers low diversity in turning trajectories, causing the vehicle to lack sufficient turning and collide with the road edge (black lines in figure). In contrast, TrajTok provides a richer set of turning trajectory tokens for the model to learn and choose from, allowing the vehicle to stay within the lane successfully.

Fig. 10 shows two vehicles moving towards each other in a parking lot. Similarly, TrajTok's vocabulary offers more diverse and drivable choices for the model to handle interactions, successfully modeling yield behavior between the two vehicles, whereas k-means leads to a collision.

## F    LLM USAGE

LLMs are used in writing for improving grammar and correcting typos.

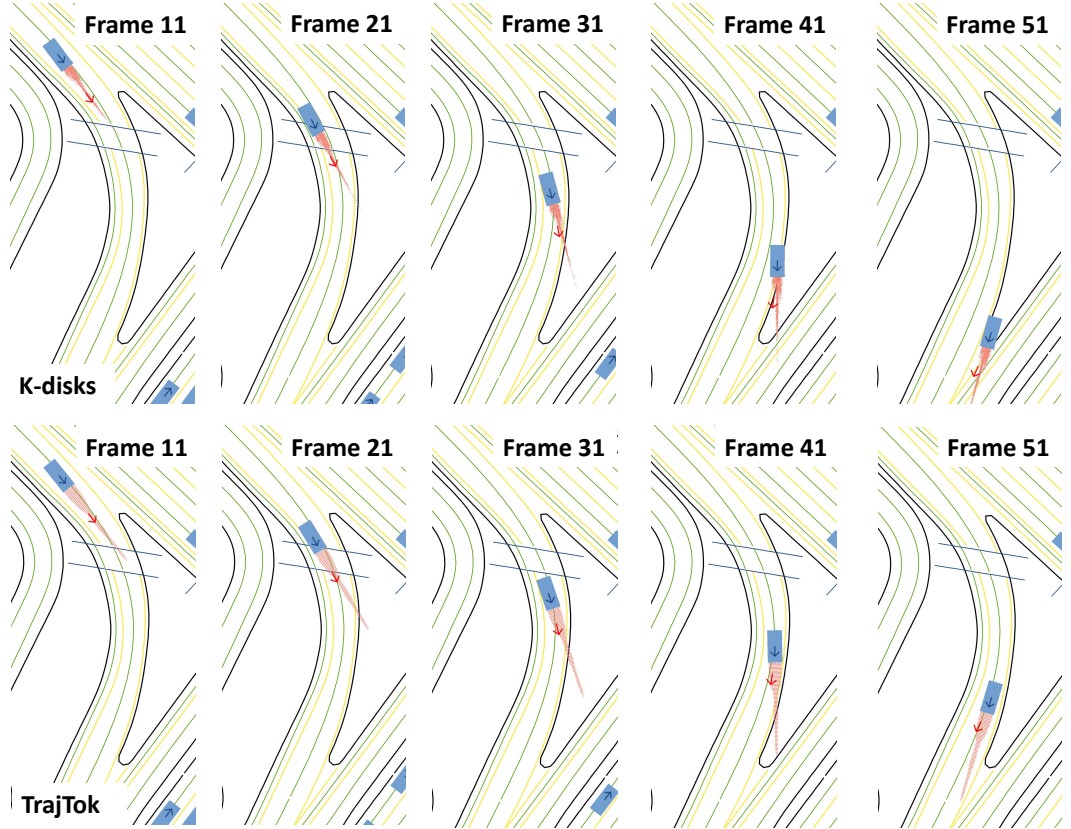

Figure 9: **Visualization in right turn.**

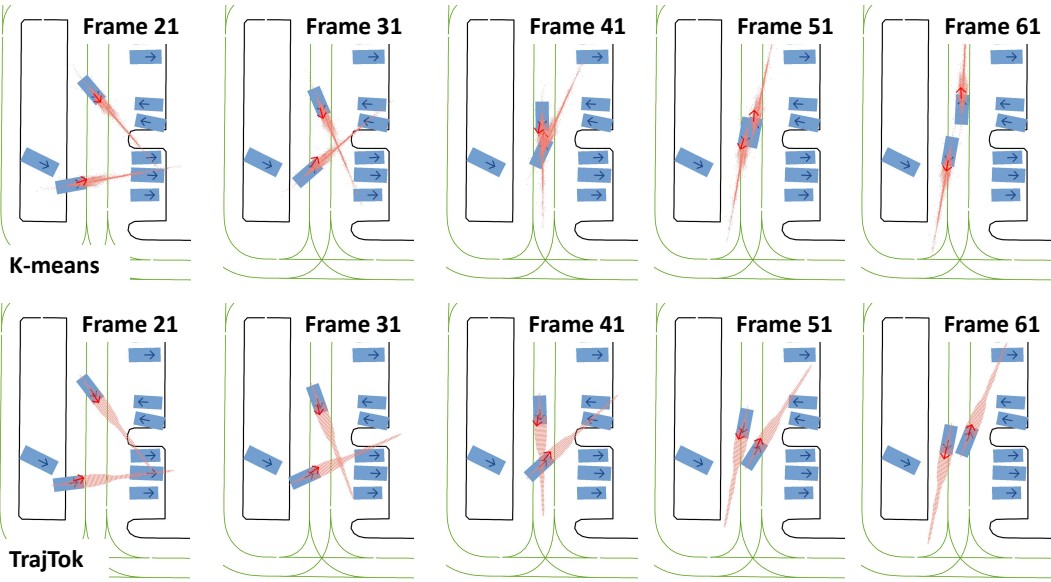

Figure 10: **Visualization at a parking lot.**

