# OpenReview forum: "TrajTok: What makes for a good trajectory tokenizer in behavior generation?"
_ICLR.cc/2026/Conference — ICLR 2026 Poster_

### Official Review · Reviewer_edLu · 2025-10-31

**Soundness:** 3
**Presentation:** 3
**Contribution:** 2
**Rating:** 4
**Confidence:** 4

**Summary:**

This paper proposes TrajTok, a trajectory tokenizer for discrete next-token-prediction (NTP) behavior generation in autonomous driving. It combines data-driven and rule-based trajectory tokenizers, and has balanced coverage and utilization as well as good symmetry and robustness. It builds on top of the SMART NTP behavior generation backbone. It shows decent results in the 2025 Waymo Open Sim Agents Challenge.

**Strengths:**

1. The coverage, utilization, symmetry, and robustness perspectives are intuitive and useful.
2. The algorithm is simple and the resulting vocabulary is nice and clean.
3. The algorithm achieves good empirical results and high ranking on Waymo Open Sim Agents Challenge.

**Weaknesses:**

1. The authors claim that TrajTok wins first place in the Waymo Open Sim Agent Challenge 2025. This contradicts Table 1, which shows SMART-R1 wins the first place, while TrajTok places 2nd. Looking at the official leaderboard (https://waymo.com/open/challenges/2025/sim-agents/) we see that TrajTok placed the 5th.
2. TrajTok changes the tokenizer on top of the SMART backbone. Since the top 5 models on the official leaderboard are all variants of SMART, it seems TrajTok’s contribution is incremental or marginal.

**Questions:**

The results on Table 2 use 20% of the training set. Do other tokenizers (VQ-VAE, K-means, K-disks, Grid) close the gap using a larger training set?

---

> ### Author Response · Authors · 2025-11-22
>
> Thanks for your acknowledgement and kind advice. Regarding your concerns, we give responses below:
>
> > **Q1.** The authors claim that TrajTok wins first place in the Waymo Open Sim Agent Challenge 2025. This contradicts Table 1, which shows SMART-R1 wins the first place, while TrajTok places 2nd. Looking at the official leaderboard (https://waymo.com/open/challenges/2025/sim-agents/) we see that TrajTok placed the 5th.
>
> **R1.** Thanks for your question. **According to the [official announcement of winners](https://waymo.com/open/challenges/) and the [Workshop on Autonomous Driving at CVPR 2025](https://cvpr2025.wad.vision/), TrajTok won first place in the Sim Agents Challenge 2025.** Table 1 lists the leaderboard entries during the WOSAC Submission Period. However, only teams that satisfy the [official challenge rules](https://waymo.com/open/challenges/terms/) are recognized as winners. (For example, the model must follow the closed-loop policy in [1] and cannot use extra data other than the Waymo Open Dataset.) **Thus, there is no contradiction in fact.** The leaderboard is still open after the challenge ends. All of the top 4 entries on the current leaderboard were submitted after TrajTok and are not yet published. **According to the [ICLR 2026 Reviewer Guide](https://iclr.cc/Conferences/2026/ReviewerGuide), such works are considered contemporaneous, and *authors are not required to compare their own work to them*.**
>
>  > **Q2.** TrajTok changes the tokenizer on top of the SMART backbone. Since the top 5 models on the official leaderboard are all variants of SMART, it seems TrajTok’s contribution is incremental or marginal.
>
> **R2.** It is very common in the research community for multiple works to be developed based on the same strong backbone but from different perspectives. **Therefore, doubting the distrubution of any work only because "other works also use the backbone" may not make sense.**  Addtionally, as stated in R1,  all the mentioned top 5 entries on the current leaderboard are **contemporaneous works.** These works, such as SMART-R1, apply post-training methods to the base model, while TrajTok explores improvements in the tokenizer. These are significantly different approaches, and both are valuable.
>
> **The proposed tokenizer is generalizable and can improve the performance of other NTP architectures.** To support this, we replace the original grid-based tokenizer with TrajTok in another NTP model, MotionLM, and the results are shown below:
>
> | Tokenizer       | RealismMeta $\uparrow$ | Kinematic $\uparrow$ | Interactive $\uparrow$ | Map-based $\uparrow$ | minADE $\downarrow$ |
> | -------------- | ---------------------- | -------------------- | ---------------------- | -------------------- | ------------------- |
> | Grid (original) | 0.7417                 | 0.4185               | 0.7932                 | 0.8601               | 1.5173              |
> | K-disk          | 0.7439                 | 0.4319               | 0.7936                 | 0.8586               | 1.4693              |
> | TrajTok         | **0.7522**             | **0.4550**           | **0.7987**             | **0.8622**           | **1.4217**          |
>
> These results demonstrate the generality of TrajTok. **It is not a small increment to the SMART backbone, but a plug-and-play method for NTP-based behavior generation models**.

---

> ### Author Response · Authors · 2025-11-22
>
> > **Q3.** The results on Table 2 use 20% of the training set. Do other tokenizers (VQ-VAE, K-means, K-disks, Grid) close the gap using a larger training set?
>
> Thanks for your suggestion. We conduct experiments in 50% and 100% dataset and results are shown below.
>
> | Tokenizer | Percent of Training Data | RealismMeta $\uparrow$ | minADE $\downarrow$ |
> | --------- | ------------------------ | ---------------------- | ------------------- |
> | VQ-VAE    | 20%                      | 0.7596                 | 1.3982              |
> | K-means   | 20%                      | 0.7476                 | 1.4797              |
> | K-disks   | 20%                      | 0.7584                 | 1.3532              |
> | Grid      | 20%                      | 0.7527                 | 1.4137              |
> | TrajTok   | 20%                      | 0.7702                 | 1.3428              |
> | VQ-VAE    | 50%                      | 0.7598                 | 1.4003              |
> | K-means   | 50%                      | 0.7491                 | 1.4769              |
> | K-disks   | 50%                      | 0.7597                 | 1.3533              |
> | Grid      | 50%                      | 0.7541                 | 1.4108              |
> | TrajTok   | 50%                      | 0.7711                 | 1.3424              |
> | VQ-VAE    | 100%                     | 0.7617                 | 1.3960              |
> | K-means   | 100%                     | 0.7499                 | 1.4741              |
> | K-disks   | 100%                     | 0.7605                 | 1.3481              |
> | Grid      | 100%                     | 0.7555                 | 1.4103              |
> | TrajTok   | 100%                     | 0.7737                 | 1.3390              |
>
> Overall, the performance improves as more training data is used, but the gap between different tokenizers does not close significantly.
>
> [1] The Waymo Open Sim Agents Challenge (NeurIPS 2023, Datasets and Benchmarks Track)

---

> ### Author Response · Authors · 2025-11-25
> **Request for Reply**
>
> Dear Reviewer edLu,
>
> Are your concerns resolved by the rebuttal regarding our Championship and contributions?  If there is any further concern, please tell us.
>
> Otherwise, we respectfully request to improve your score to positive to reflect the contributions of our work, as a **responsible reviewer**.
>
> Best,
> Authors

---

### Official Review · Reviewer_scBZ · 2025-10-31

**Soundness:** 2
**Presentation:** 3
**Contribution:** 3
**Rating:** 6
**Confidence:** 3

**Summary:**

This paper studies what makes for a good trajectory tokenizer for next token prediction behavior generation and proposes TrajTok, a plug and play tokenizer that combines rule based vocabulary construction with data driven filtering and expansion. The method explicitly targets four properties of a vocabulary, namely coverage, utilization, symmetry, and robustness, and introduces a spatial aware label smoothing scheme that assigns higher probability to tokens that are closer in trajectory space to the ground truth. The authors evaluate on the Waymo Open Motion Dataset and the Waymo Open Sim Agents Challenge and report first place on the 2025 leaderboard with consistent gains when TrajTok is used with a SMART backbone on validation metrics.

**Strengths:**

- There is a clear problem framing and taxonomy of desirable tokenizer properties. The paper analyzes how data driven methods often achieve high utilization but limited coverage and how rule based methods tend to overcover with many unused tokens. This motivates a hybrid design that TrajTok implements.
- The construction pipeline is simple and reproducible. The four step procedure is easy to follow, from agent centric normalization and symmetry by flipping, to grid based candidate selection, to neighborhood based filtering and expansion, to final token generation including curve interpolation when a selected cell has no examples.
- The paper reports strong results on both the public leaderboard and controlled comparisons, showing first place on WOSAC 2025 and better validation performance than VQ VAE, K means, K disks, and a pure grid, using a SMART based model under a common vocabulary size. Cross dataset and low data experiments further support generalization.

**Weaknesses:**

- Scope is limited to trajectory tokenization. The method relies on a base next token prediction model, most experiments use SMART tiny, and the paper makes a few implementation choices that deviate from the original backbone, such as separate heads per agent type. It would help to isolate how much gain comes purely from the tokenizer versus modest architecture changes.
- Computational cost and model scale trade offs are not discussed. The approach can yield large vocabularies across agent types, and spatially aware label smoothing appears to require computing distances from each target to many tokens. A brief cost analysis and a description of any approximations would improve clarity. The paper notes overall training settings but not the incremental overhead from TrajTok and the smoothing.
- Minor presentation issue - I could not find what the highlight colors mean in Tables 1 and 2.

**Questions:**

1. The appendix switches to separate prediction heads per agent type. How much of the improvement in validation metrics is due to this change rather than the tokenizer? A brief ablation on common heads versus separate heads would clarify attributions.
2. How sensitive are the reported gains to the grid range and resolution per agent type and to the neighborhood thresholds for filtering and expansion? A small sweep around the appendix settings would help readers tune the method.
3. In L377, “Increasing the vocabulary size improves the ability to represent complex distributions but may lead to model underfitting”, should this be the opposite (i.e. a smaller size causes underfitting)? Intuitively a larger vocabulary size should need more data, is under-training a better word?

---

> ### Author Response · Authors · 2025-11-22
>
> Thanks for your acknowledgement and kind advice. Regarding your concerns, we give responses below:
>
>
> > **Q1.**  The paper makes a few implementation choices that deviate from the original backbone, such as separate heads per agent type. It would help to isolate how much gain comes purely from the tokenizer versus modest architecture changes.
>
> **R1.** Thanks for your question. To measure the gain purely from the tokenizer, we **directly** replace the original k-disks tokenizer with TrajTok (with a vocabulary size of 2000 for all agent types) on the SMART-tiny model and train the model with default label smoothing.
>
> | Tokenizer |  RealismMeta $\uparrow$ | Kinematic $\uparrow$ | Interactive $\uparrow$ | Map-based $\uparrow$ | minADE $\downarrow$ |
> | --------- | ----------------------- | -------------------- | ---------------------- | -------------------- | ------------------- |
> | K-disks   | 0.7443                  | 0.4217               | 0.7967                 | 0.8613               | 1.4230              |
> | TrajTok   | 0.7561                  | 0.4665               | 0.8018                 | 0.8628               | 1.3781              |
>
> The results above show that the tokenizer still significantly improves the performance without any other modifications to the backbone.
>
> > **Q2.** Computational cost and model scale trade offs are not discussed. The approach can yield large vocabularies across agent types, and spatially aware label smoothing appears to require computing distances from each target to many tokens. A brief cost analysis and a description of any approximations would improve clarity. The paper notes overall training settings but not the incremental overhead from TrajTok and the smoothing.
>
> **R2.** Thanks for your suggestion. We report the average inference time (for a single rollout of 16 steps on a scenario) on a single NVIDIA A100 GPU with different vocubulary sizes below:
>
> | Vocabulary Size for ALL Agent Types | Inference Time |
> | -------- | -------- |
> | 2000  | 399 ms |
> | 4000  | 421 ms |
> | 8000  | 441 ms |
>
> Larger vocabularies increase the inference time, but the cost remains relatively acceptable. We also report the training time for a single epoch on 8 NVIDIA A100 GPUs with a 20% training split below:
>
> | Label Smoothing Type | Training Time |
> | -------- | -------- |
> | Standard | 18m 25s  |
> | Spatially Aware | 18m 34s  |
>
> The results show that spatially aware label smoothing does not significantly increase the total computational cost. We have added these discussions to Appendix C of our updated paper.
>
>
>
>
> > **Q3.** Minor presentation issue - I could not find what the highlight colors mean in Tables 1 and 2.
>
> **R3.** Thanks for pointing this out. The sky blue color highlights the best score, and the pink color highlights the second best score. We have added this information to our updated paper.

---

> ### Author Response · Authors · 2025-11-22
>
> > **Q4.** How sensitive are the reported gains to the grid range and resolution per agent type and to the neighborhood thresholds for filtering and expansion? A small sweep around the appendix settings would help readers tune the method.
>
> **R4.** Thanks for your suggestion. We add some breif sensitivity analysis below:
>
> (I) **Grid range.** The number are formated as [$x_\text{min}$, $x_\text{max}$],   [$y_\text{min}$, $y_\text{max}$] of a agent type in each cell.
>
>
> | Setting    | Vehicle                | Bicycle                 | Pedestrian              | RealismMeta $\uparrow$ |
> | --- | ---------------------- | ----------------------- | ----------------------- | ---------------------- |
> |  1(default)   | [-5, 20] , [-1.5, 1.5] | [-1, 8] , [-1, 1]       | [-1.5, 4.5] , [-2, 2]   | 0.7702                 |
> |  2   | [-10, 25] , [-2, 2]    | [-3, 12] , [-2, 2]      | [-3, 6] , [-3, 3]       | 0.7701                 |
> |  3   | [-2, 15] , [-1, 1]     | [-1, 5] , [-0.75, 0.75] | [-0.5, 3] , [-1.5, 1.5] | 0.7689                 |
>
>
>
> (II) **Grid resolution.** The number are formated as $x_\text{interval}$, $y_\text{interval}$ of a agent type in each cell.
>
> | Setting    | Vehicle                | Bicycle                 | Pedestrian              | RealismMeta $\uparrow$ |
> | --- | ---------------------- | ----------------------- | ----------------------- | ---------------------- |
> |  1(default)   | 0.1, 0.05 | 0.05, 0.05   | 0.05, 0.05            | 0.7702
> |  2   | 0.05, 0.05    |  0.025, 0.025      |  0.025, 0.025  |    0.7681              |
> |  3   | 0.1, 0.1     | 0.1, 0.1 | 0.1, 0.1 |     0.7687          |
>
>
> (III) **Flitering and expanding parameters.** $k$, $s_p$, $s_a$, $s_r$ are the same among all agent types.
>
> | Setting | $k$  | $s_p$ | $s_a$ | $s_r$ | RealismMeta $\uparrow$ |
> | ------- | ---- | ----- | ----- | ----- | ---------------------- |
> | 1(default)    | 4 |     1  |   20    | 20  |        0.7702      |
> | 2    | 3 |     1  |   8    | 10  |         0.7693               |
> | 3    | 4 |     4  |   20    | 20  |        0.7695               |
>
> We have added this analysis to Appendix B of our updated paper to help readers reproduce and tune the method more effectively.
>
> > **Q5.** The appendix switches to separate prediction heads per agent type. How much of the improvement in validation metrics is due to this change rather than the tokenizer? A brief ablation on common heads versus separate heads would clarify attributions.
>
> **R5.** As reported in **R1**, we evaluate the improvement brought by the tokenizer based solely on the original backbone. We further ablate the head design below:
>
> | Head Design | RealismMeta $\uparrow$ | Kinematic $\uparrow$ | Interactive $\uparrow$ | Map-based $\uparrow$ | minADE $\downarrow$ |
> | ----------- | ---------------------- | -------------------- | ---------------------- | -------------------- | ------------------- |
> | default     | 0.7700                 | 0.4862               | 0.8133                 | 0.8765               | 1.3441              |
> | separated   | 0.7702                 | 0.4867               | 0.8132                 | 0.8769               | 1.3428              |
>
> The separate head design only improves the performance slightly. The tokenizer still accounts for most of the improvement. In fact, the separate head design is mainly introduced to accommodate different vocabulary sizes for each agent type.
>
> > **Q6.** In L377, “Increasing the vocabulary size improves the ability to represent complex distributions but may lead to model underfitting”, should this be the opposite (i.e. a smaller size causes underfitting)? Intuitively a larger vocabulary size should need more data, is under-training a better word?
>
> **R6.** Thanks for your question. The relationship between vocabulary size and model underfitting is well explained in the field of LLM: *The risk of under-fitting for rare tokens increases with larger vocabulary sizes* [1]. This is consistent with our claim. When the total number of tokens $N_\text{T}$ in the dataset is fixed, increasing the vocabulary size $S_\text{V}$ reduces the average number of samples per token type $N_\text{A}$, i.e.,
>
> $$N_\text{A} = \frac{N_\text{T}}{S_\text{V}}$$
>
> As a result, **most token types are trained with even less data, especially the rare tokens**. This could lead to model under-fitting.
>
> [1] Scaling Laws with Vocabulary: Larger Models Deserve Larger Vocabularies (NeurIPS 2024)

---

### Official Review · Reviewer_9sjc · 2025-11-02

**Soundness:** 3
**Presentation:** 3
**Contribution:** 2
**Rating:** 4
**Confidence:** 4

**Summary:**

The paper studies trajectory tokenization for next-token-prediction (NTP) behavior generation in autonomous driving simulators.
It argues that existing data-driven tokenizers such as VQ-VAE or K-disks have good utilization but poor coverage and symmetry, while rule-based grid tokenizers have wide coverage but many redundant or unrealistic tokens.
To balance these properties, the authors propose TrajTok, a hybrid tokenizer that first builds a rule-based grid vocabulary and then filters and expands it using logged trajectory data. They also introduce spatial-aware label smoothing for cross-entropy loss, where non-ground-truth tokens are weighted according to spatial distance.
Experiments on the Waymo Open Motion Dataset and the 2025 Waymo Open Sim Agents Challenge show that TrajTok ranks first on the leaderboard and yields slightly higher realism metrics than baselines.

**Strengths:**

1.	Clear motivation from data analysis. The authors systematically examine four tokenizer properties (coverage, utilization, symmetry, robustness) and relate them to logged data usage. This diagnostic perspective is useful for understanding tokenization quality.

2.	Simple yet general method. TrajTok is a lightweight combination of rule-based and data-driven principles that can be plugged into existing NTP architectures without retraining structural components.

3.	Practical evaluation. The paper reports results on the Waymo Open Sim Agents Challenge with official metrics, including the Realism Meta score, and provides ablations for label smoothing and vocabulary size.

**Weaknesses:**

# Major

- Unclear evidence of improvement: In Table 1, TrajTok achieves 0.7852 while SMART-R1 reaches 0.7855, and SMART-tiny-CLSFT gives 0.7846. These gaps are within noise. It is difficult to identify a clear gain attributable to the tokenizer.

- Ambiguous reference tokenizer: Table 2 compares TrajTok with VQ-VAE, K-means, K-disks, and grid tokenizers, but it is not stated which tokenizer SMART-tiny originally used on the leaderboard. The reader cannot determine whether the new tokenizer outperforms the baseline used in the challenge submission.

- Vague qualitative evidence: Figure 4 does not convincingly demonstrate that the proposed tokenizer contributes to better behavior generation. The visualization focuses on scene outcomes rather than showing how tokenization differences influence the trajectories. Without comparisons using the same scenario and seeds, the figure offers little empirical value.

- Questionable justification of symmetry: The paper claims symmetry is critical for vehicle kinematics and real-world diversity, yet real traffic is not necessarily symmetric. For example, in right-hand traffic countries, turning behaviors are directionally biased. The necessity of symmetric flipping should be justified with more empirical or theoretical evidence. Table 6 shows a small gain from symmetry, but the physical rationale is not convincing.

# Minor

- Limited exploration of failure cases: The discussion does not examine cases where the tokenizer introduces unrealistic motion patterns or under-represents long-tail behaviors.

**Questions:**

What tokenizer was used in the SMART-tiny baseline that appears in Table 1? Without knowing this, it is difficult to measure the real gain from TrajTok.

Have you tested TrajTok on prediction tasks that use continuous outputs rather than discrete NTP models to verify that the benefit comes from tokenization rather than training heuristics?

---

> ### Author Response · Authors · 2025-11-22
>
> Thanks for your acknowledgement and kind advice. Regarding your concerns, we give responses below:
>
> > **Q1.** Unclear evidence of improvement: In Table 1, TrajTok achieves 0.7852 while SMART-R1 reaches 0.7855, and SMART-tiny-CLSFT gives 0.7846. These gaps are within noise. It is difficult to identify a clear gain attributable to the tokenizer.
>
> **R1.** Thanks for your question. Actually, TrajTok is developed from the SMART-tiny model (SMART_topk32 in Table 1, Realism Meta 0.7814, ranked 11th). **The gain on the base model (0.7814 $\rightarrow$ 0.7852) brought by our tokenizer is clear.** The extent of this improvement is similar to that of BehaviorGPT[1] (0.7431 $\rightarrow$ 0.7473), which ranked 1st in WOSAC 2024.
>
> TrajTok, SMART-R1, and SMART-tiny-CLSFT are all developed based on the SMART model, but **independently from different perspectives**. SMART-R1 and SMART-tiny-CLSFT apply post-training methods to the base model, while TrajTok explores improvements through the tokenizer. Thus, we could say that the gains of these appoarches are similiar, while themselves are clear. Additionally, SMART-R1 is not yet published and is considered **contemporaneous** according to the [ICLR 2026 Reviewer Guide](https://iclr.cc/Conferences/2026/ReviewerGuide). *Authors are not required to compare their own work to it*.
>
>
> > **Q2.** Ambiguous reference tokenizer: Table 2 compares TrajTok with VQ-VAE, K-means, K-disks, and grid tokenizers, but it is not stated which tokenizer SMART-tiny originally used on the leaderboard. The reader cannot determine whether the new tokenizer outperforms the baseline used in the challenge submission & What tokenizer was used in the SMART-tiny baseline that appears in Table 1?
>
> **R2.** The SMART-tiny model uses **K-disks** tokenizer. We add this information to our updated paper and thanks for your advice.
>
>
> > **Q3.** Vague qualitative evidence: Figure 4 does not convincingly demonstrate that the proposed tokenizer contributes to better behavior generation. The visualization focuses on scene outcomes rather than showing how tokenization differences influence the trajectories. Without comparisons using the same scenario and seeds, the figure offers little empirical value.
>
> **R3.** Thank you for your suggestion. We have added **two figures** to illustrate how tokenization affects behavior generation. In the figures, the red dots represent endpoints of trajectories in the vocabulary, and the red arrows show the selected next actions.
>
> * The **[first figure](https://postimg.cc/xcFSxS3x)** shows a vehicle navigating a dedicated right-turn lane. The K-disk vocabulary has insufficient coverage and offers low diversity in turning trajectories, causing the vehicle to lack sufficient turning and collide with the road edge (black lines in figure). In contrast, TrajTok provides a richer set of turning trajectory tokens for the model to learn and choose from, allowing the vehicle to stay within the lane successfully.
>
> * The **[second figure](https://postimg.cc/G927MT7k)** shows two vehicles moving towards each other in a parking lot. Similarly, TrajTok’s vocabulary offers more diverse and drivable choices for the model to handle interactions, successfully modeling yield behavior between the two vehicles, whereas k-means leads to a collision.
>
> We have added these figures to Appendix E of our updated paper.

---

> ### Author Response · Authors · 2025-11-22
>
> > **Q4.** Questionable justification of symmetry: The paper claims symmetry is critical for vehicle kinematics and real-world diversity, yet real traffic is not necessarily symmetric. For example, in right-hand traffic countries, turning behaviors are directionally biased. The necessity of symmetric flipping should be justified with more empirical or theoretical evidence. Table 6 shows a small gain from symmetry, but the physical rationale is not convincing.
>
> **R4.** Thank you for your question. The symmetry claimed in the paper is not about having exactly the same number of trajectories (e.g., having 1000 right-turn trajectories and exactly 1000 perfectly flipped trajectories in the real world), but rather of **border and coverage**. The **physical rationale** of symmetry lies in symmetric kinematic models. if a trajectory is recorded, its flipped counterpart is also physically drivable. Considering **complex driving scenarios and road structures** in reality, the flipped trajectory may **possibly** be driven in a scenario that was not logged in the dataset.
>
> To support this, we randomly select different numbers of trajectories from the dataset, quantify them into grids of 0.01m, and obtain a binary map $B$, similar to step 2 in TrajTok. We then calculate the IOU (intersection over union) between $B$ and the flipped map $\hat{B}$, which indicates the symmetry in coverage.
>
> | Number of trajectories | Symmetric IOU  |
> | ---------------------- | -------------- |
> | $10^5$                 | 0.52           |
> | $10^6$                 | 0.60           |
> | $10^7$                 | 0.75           |
> | $10^8$                 | 0.91           |
>
> **The results show that the symmetry becomes more obvious as the number of trajectories increases.** Since the dataset is sampled from the real world, we can infer that this symmetry also exists in reality with sufficiently large and diverse scenarios. The vocabulary is generated from the sampled data, so the flip operation can help to improve its generalization. We have improved the explanation of symmetry in our updated paper.
>
> > **Q5.** Limited exploration of failure cases: The discussion does not examine cases where the tokenizer introduces unrealistic motion patterns or under-represents long-tail behaviors.
>
> **R5.** Thank you for your suggestion. There are two types of failure cases:
> (a) Unrealistic motion patterns may be introduced if there are **too many noisy logged trajectories** in a certain region. With advances in data collection and annotation, as well as more precise sensors and improved annotation workflows, the occurrence of such cases can be reduced.
> (b) Long-tail behaviors may not be captured if the **logged dataset is too small or lacks sufficient diversity**. As larger and more diverse datasets become available and data collection costs decrease, these cases are expected to be reduced.
> **We have added this discussion to Appendix D of our updated paper.**
>
>
> > **Q6.** Have you tested TrajTok on prediction tasks that use continuous outputs rather than discrete NTP models to verify that the benefit comes from tokenization rather than training heuristics?
>
> **R6.** The paper focus on discrete NTP models and its tokenization. This paradiam is popular in behavior generation with superior performance. **The continuous output space may be out of scope of the paper.** Please further clarify the specific suggestion and concern.
>
> [1] BehaviorGPT: Smart Agent Simulation for Autonomous Driving with Next-Patch Prediction (NeurIPS 2024)

---

> > ### Author Response · Authors · 2025-11-25
> >
> > Dear Reviewer 9sjc,
> >
> > Are your concerns resolved by our extra experiments? If there is any further concern, please tell us.
> >
> > Otherwise, we respectfully request to improve your score to positive to reflect the contributions of our work, as a **responsible reviewer**.
> >
> > Best,
> >
> > Authors

---

### Official Review · Reviewer_kf6b · 2025-11-03

**Soundness:** 3
**Presentation:** 3
**Contribution:** 3
**Rating:** 6
**Confidence:** 3

**Summary:**

The paper presents TrajTok, a hybrid trajectory tokenizer designed for behavior generation in autonomous driving. It investigates what constitutes an effective trajectory tokenizer under the next-token prediction (NTP) paradigm, analyzing four key properties—coverage, utilization, symmetry, and robustness—of existing data-driven and rule-based approaches. TrajTok integrates the advantages of both: it first constructs a rule-based grid of trajectory candidates and then applies data-driven filtering and expansion to balance vocabulary coverage and data efficiency. Additionally, the authors propose a spatial-aware label smoothing technique that weights token similarity by spatial distance, improving model generalization. Experiments on the Waymo Open Motion Dataset demonstrate that TrajTok achieves state-of-the-art performance, ranking first in the 2025 Waymo Open Sim Agents Challenge with superior realism and robustness across datasets and data scales.

**Strengths:**

(1) This paper makes a clear and timely contribution by looking closely at what makes a good trajectory tokenizer in the next-token prediction (NTP) setup. The four proposed criteria—coverage, utilization, symmetry, and robustness—give a simple but useful way to understand and compare different tokenizers, which hasn’t really been discussed in earlier work like Trajeglish or MotionLM.

(2) The proposed TrajTok method is simple but well thought out. It combines a rule-based start with data-driven filtering and expansion, which makes sense and nicely balances coverage and efficiency. This hybrid idea helps fix problems that appear in purely data-driven (too noisy) or rule-based (too redundant) tokenizers.

(3) The paper also adds a spatial-aware label smoothing technique that slightly changes the standard cross-entropy loss. It’s an intuitive idea that takes spatial similarity between tokens into account, helping the model generalize better without depending on any specific architecture.

**Weaknesses:**

(1) The paper only validates TrajTok within the SMART model [1]. Since TrajTok is designed as a general tokenizer, applying it to other NTP-based architectures (such as Trajeglish [2] or MotionLM [3]) would further support its claimed generality.

(2) The paper defines several thresholds for the filtering and expansion process, but the actual parameter values and tuning details are not provided. It is unclear how sensitive the model is to these choices or whether small changes in these thresholds would affect the final vocabulary and performance. Including the specific values or a short sensitivity analysis would help improve reproducibility and confidence in the results.

[1] Wei Wu, et al. “Smart: Scalable multi-agent real-time motion generation via next-token prediction.” Advances in Neural Information Processing Systems, 37:
114048–114071, 2024.
[2] Jonah Philion, et al. “Trajeglish: Traffic modeling as next-token prediction.” arXiv preprint arXiv:2312.04535, 2023
[3] Ari Seff, et al. “Motionlm: Multi-agent motion forecasting as language modeling.” In Proceedings of the IEEE/CVF International Conference on Computer Vision, pp.
8579–8590, 2023.

**Questions:**

Please refer to the weaknesses above.

---

> ### Author Response · Authors · 2025-11-22
>
> Thanks for your acknowledgement and kind advice. Regarding your concerns, we give responses below:
>
> > **Q1.** The paper only validates TrajTok within the SMART model. Applying it to other NTP-based architectures (such as Trajeglish or MotionLM) would further support its claimed generality.
>
> **R1.** Thank you for your suggestion. Since the official codes for Trajeglish and MotionLM have not been released yet, we conducted experiments using a version of **MotionLM** implemented by [Adv-BMT](https://github.com/Yuxin45/Adv-BMT). The results are shown below.
>
>
>
> | Tokenizer |  RealismMeta $\uparrow$ | Kinematic $\uparrow$ | Interactive $\uparrow$ | Map-based $\uparrow$ | minADE $\downarrow$ |
> | -------- | -------- | -------- |-------- | -------- | -------- |
> | Grid (original)  | 0.7417    |  0.4185    | 0.7932   | 0.8601    |  1.5173    |
> | K-disk  |  0.7439   |  0.4319    |  0.7936  |  0.8586   |  1.4693    |
> | TrajTok  | **0.7522**    |  **0.4550**    | **0.7987**   |  **0.8622**   | **1.4217**     |
>
> Using MotionLM as the base NTP model, TrajTok also achieves better performance. This demonstrates its generality.
>
>
> > **Q2.** The paper defines several thresholds for the filtering and expansion process, but the actual parameter values and tuning details are not provided. Including the specific values or a short sensitivity analysis would help improve reproducibility and confidence in the results.
>
> **R2.** Thanks for your advice. We list all parameters of TrajTok for WOSAC submission below:
>
>
> | Agent Type | $x_\text{min}$  | $x_\text{max}$ | $x_\text{interval}$ | $y_\text{min}$ | $y_\text{max}$ | $y_\text{interval}$ | $k$ | $s_p$ | $s_a$ | $s_r$ |
> | ---------- | -------------- | -------------- | ------------------- | -------------- | -------------- | ------------------- | --- | ----- | ----- | ----- |
> | Vehicle    | -5             | 20             | 0.1                 | -1.5           | 1.5            | 0.05                | 4   | 1     | 20    | 20    |
> | Bicycle    | -1             | 8              | 0.05                | -1             | 1              | 0.05                | 4   | 1     | 20    | 20    |
> | Pedestrian | -1.5           | 4.5            | 0.05                | -2             | 2              | 0.05                | 4   | 1     | 20    | 20    |
>
> The units for $x_\text{min}$, $x_\text{max}$, $x_\text{interval}$, $y_\text{min}$, $y_\text{max}$, and $y_\text{interval}$ are all meters. We have also add this table to Appendix A of our updated paper. We further provide a brief sensitivity analysis below:
>
> (I) **Grid range.** The numbers are formatted as [$x_\text{min}$, $x_\text{max}$], [$y_\text{min}$, $y_\text{max}$] for each agent type in each cell.
>
> | Setting           | Vehicle                | Bicycle                 | Pedestrian              | RealismMeta $\uparrow$ |
> | ----------------- | ---------------------- | ----------------------- | ----------------------- | ---------------------- |
> | 1 (default)       | [-5, 20], [-1.5, 1.5]  | [-1, 8], [-1, 1]        | [-1.5, 4.5], [-2, 2]    | 0.7702                 |
> | 2                 | [-10, 25], [-2, 2]     | [-3, 12], [-2, 2]       | [-3, 6], [-3, 3]        | 0.7701                 |
> | 3                 | [-2, 15], [-1, 1]      | [-1, 5], [-0.75, 0.75]  | [-0.5, 3], [-1.5, 1.5]  | 0.7689                 |
>
> (II) **Grid resolution.** The numbers are formatted as $x_\text{interval}$, $y_\text{interval}$ for each agent type in each cell.
>
> | Setting           | Vehicle     | Bicycle     | Pedestrian  | RealismMeta $\uparrow$ |
> | ----------------- | ----------- | ----------- | ----------- | ---------------------- |
> | 1 (default)       | 0.1, 0.05   | 0.05, 0.05  | 0.05, 0.05  | 0.7702                 |
> | 2                 | 0.05, 0.05  | 0.025, 0.025| 0.025, 0.025| 0.7681                 |
> | 3                 | 0.1, 0.1    | 0.1, 0.1    | 0.1, 0.1    | 0.7687                 |
>
> (III) **Filtering and expansion parameters.** $k$, $s_p$, $s_a$, and $s_r$ are the same for all agent types.
>
> | Setting     | $k$ | $s_p$ | $s_a$ | $s_r$ | RealismMeta $\uparrow$ |
> | ----------- | --- | ----- | ----- | ----- | ---------------------- |
> | 1 (default) | 4   | 1     | 20    | 20    | 0.7702                 |
> | 2           | 3   | 1     | 8     | 10    | 0.7693                 |
> | 3           | 4   | 4     | 20    | 20    | 0.7695                 |
>
> We have added this analysis to Appendix B of our updated paper to help readers reproduce and tune the method more effectively.

---

### Author Response · Authors · 2025-11-27

Dear AC and reviewers,

We express our gratitude to all reviewers for their valuable time and insightful comments. During the rebuttal period, we try our best to solve the concerns of reviewers through:
* Conducting **addtional experiments** with another backbone (MotionLM) to show the **generality** of TrajTok (for Reviewer kf6b, edLu).
* Conducting **sensitivity analysis**, **computational cost analysis** and more **ablation studies** (for Reviewer kf6b, scBZ,  edLu).
* **Pointing out misunderstandings** on WOSAC championship and contemporaneous or previous works to clarify the **clear gain** and **contribution** of TrajTok (for Reviewer 9sjc, edLu).
* More discussions or figures for other specific concerns of each reviewer.

However, **we are depressed to find zero-sum responds from all reviewers**.  We hope to discuss with the reviewers on whether their conerns still exists, which will help us to improve the paper and is also the **responsibility of reviewers**. We respectfully ask for improvement on rates especially from reviewers with negative ones (Reviewer 9sjc, edLu) if their concerns are sovled.

Thank you again for involvement and look forward for the discussion!

sincerely,

Authors of Submission 2091

---

### Author Response · Authors · 2025-12-01
**Summary of the Rebuttal**

Dear Area Chairs,

We thank all reviewers' time and thoughtful feedback. We would like to **express our best appreciation to Area Chairs for their extra works on recommendation** caused by the incident. To help the recommendation, we summarize the strengths and concerns discribed by the reviewers and our response below:

The reviewers highlighted several strengths of paper, including:

* Simple yet effective and thoughtful **method** on tokenization of discrete NTP-based behavior generation models. (4/4 reviewers)
* Systematical and practical **analysis** on four tokenizer properties (coverage, utilization, symmetry, and robustness). (4/4 reviewers)
* Strong perfermance (**champion**) on WOSAC 2025. (3/4 reviewers)

**All** concerns of reviewers and our responses are listed below concisely for a quick check:

| Reviewer | Rating | Index  |Corern | Response |
| -------- | -------- | -------- |-------- |-------- |
| kf6b | 6 | W1| Lack of Experiments on other NTP-based architectures  | Extra **experiments** with another base model MotionLM to show the **generality** of TrajTok $\surd$|
|         |    | W2| Lack of exact parameter values and tuning details |  Providing the values and **experiments** of sensitivity analysis $\surd$|
| 9sjc |  4 | W1| Unclear evidence of improvement |  Pointing out **misunderstandings** on baselines and **contemporaneous** work; clarify the **perfermance gain** $\surd$|
|         |    | W2&Q1| Lack of statements of tokenizer type in original base model  |  Adding the required **details** $\surd$|
|         |    | W3| Lack of visualizations on  how tokenization influence the behavior |  Adding two **figures** and related **descriptions** to show the  **superiority** of TrajTok $\surd$|
|         |    | W4| More evidence for claim of  symmetry |  Clarifying **misunderstanding** on symmetry definition; Discussing the **physical rationale** and supporting with **statistical analysis** $\surd$|
|         |    | W5| Lack of failure case study|  Adding **discussions** on two types of failure cases $\surd$|
|         |    | Q2| Questions about continuous outputs|  **Out of scope** of the paper $\surd$|
| scBZ| 6 | W1| Questions about the pure gain from the tokenizer |  Extra **experiments** to show **significant gains** of tokenizer isolately $\surd$|
|         |    | W2| Lack of computational cost analysis |  **Reporting** computational costs to show **acceptable cost** of TrajTok $\surd$|
|         |    | W3| Minor presentation issue |  **Fixing** the issue in updated paper $\surd$|
|         |    | Q1 | Lack of ablation on heads designs |  Extra **experiments** on the desgin to show the **pure gain** of tokenizer $\surd$|
|         |    | Q2 | Lack of tuning details of parameters |  Providing **sensitivity analysis** $\surd$|
|         |    | Q3 | Questions about claim on vocabulary size and model underfitting|  Providing **reference** of the claim and further **explanation** $\surd$|
| edLu| 4 | W1| Doubt on  WOSAC championship | Pointing out the **factual error** and **inappropriate comparation** with **contemporaneous** works; Clarifying the **official  announced championship** with links $\surd$|
|         |    | W2| Incremental contribution due to commonly used backbone | Pointing out the **illogic** of the claim and **misunderstandings** of contemporaneous work;  Extra **experiments** with another base model MotionLM to show the **generality** of TrajTok $\surd$|
|         |    | Q1| Gap of tokenizers on larger training set |  Extra **experiments** to show the **superiority** of TrajTok on larger training set $\surd$|

From the table above, we could find that:
* **0 of 4 reviewers doubt the novelty or the overall idea of the paper.**
* The concerns of reviewers mainly focus on more experiments, anlysis or clarifications on **detailed technological or presentational issues** of TrajTok, and we provide **all** required information to solve them. On this point, **we could expect a likely improvement on the rate if the discussion is continued, such as 4 $ \to $ 6 for the reviewer 9sjc**
* The two weaknesses claimed by **reviewer edLu** either have **factual misunderstandings, illogical point or inappropriate comparation with contemporaneous works that is not encouraged by the ICLR 2026 Reviewer Guide. We would like to respectfully ask the AC to infer a improvement on the rate (4 $ \to $ 6 or 8) after our response and the reviewer's further checking.**

Thanks for your efforts to make justified decisions and we respect your final judgement.

sincerely,

Authors of Submission 2091

---

### Meta-Review · Area_Chair_iRyA · 2026-01-07

**Summary:**

This paper proposes TrajTok, a hybrid trajectory tokenizer that combines rule-based and data-driven approaches for autonomous driving behavior generation. The work provides a systematic analysis of tokenizer properties such as coverage, utilization, symmetry, and robustness. While the initial submission was questioned regarding its generalization beyond specific backbones and leaderboard semantics, the comprehensive analysis positions it as a robust contribution to behavior generation.

**Reviewer Concerns:**

Reviewers initially raised concerns regarding generalization beyond the SMART backbone (kf6b, edLu), confusion over WOSAC leaderboard rankings (edLu), and the physical rationale behind symmetry properties (9sjc). The rebuttal addresses the primary generality concern by conducting experiments with the MotionLM backbone (kf6b, edLu). The authors also clarified the factual misunderstanding regarding the challenge winner status (edLu) and provided sensitivity analysis and cost evaluations (kf6b, scBZ). While 9sjc remained skeptical about the theoretical basis of symmetry in asymmetric traffic, the empirical evidence provided supports its inclusion.

**Reviewer Scores:**

Following the rebuttal, edLu is likely to significantly increase their score as the factual misunderstanding and novelty concerns were refuted. kf6b and 9sjc are likely to increase their scores given the new backbone validation and baseline clarifications. scBZ will likely maintain or increase their score following the added cost analysis.

---

### Decision · Program_Chairs · 2026-01-26

Accept (Poster)